# Ciliomotor circuitry underlying whole-body coordination of ciliary activity in the *Platynereis* larva

**Csaba Verasztó, Nobuo Ueda, Luis A Bezares-Calderón, Aurora Panzera, Elizabeth A Williams, Réza Shahidi, Gáspár Jékely\***

Max Planck Institute for Developmental Biology, Tübingen, Germany

**Abstract** Ciliated surfaces harbouring synchronously beating cilia can generate fluid flow or drive locomotion. In ciliary swimmers, ciliary beating, arrests, and changes in beat frequency are often coordinated across extended or discontinuous surfaces. To understand how such coordination is achieved, we studied the ciliated larvae of *Platynereis dumerilii*, a marine annelid. *Platynereis* larvae have segmental multiciliated cells that regularly display spontaneous coordinated ciliary arrests. We used whole-body connectomics, activity imaging, transgenesis, and neuron ablation to characterize the ciliomotor circuitry. We identified cholinergic, serotonergic, and catecholaminergic ciliomotor neurons. The synchronous rhythmic activation of cholinergic cells drives the coordinated arrests of all cilia. The serotonergic cells are active when cilia are beating. Serotonin inhibits the cholinergic rhythm, and increases ciliary beat frequency. Based on their connectivity and alternating activity, the catecholaminergic cells may generate the rhythm. The ciliomotor circuitry thus constitutes a stop-and-go pacemaker system for the whole-body coordination of ciliary locomotion.

**\*For correspondence:** gaspar.
jekely@tuebingen.mpg.de

**Competing interests:** The authors declare that no competing interests exist.

## Introduction

Multiciliated surfaces characterised by many beating cilia are widespread in eukaryotes. Such surfaces can effectively generate flow in many different contexts, including the driving of solute transport in reef corals (*Shapiro et al., 2014*), moving mucus and particles in mucociliary epithelia (*Kramer-Zucker et al., 2005*; *Walentek et al., 2014*), driving the cerebrospinal fluid (*Faubel et al., 2016*), or moving the ovum in the mammalian oviduct (*Halbert et al., 1989*). Multiciliated surfaces can also drive locomotion, as in ciliates, colonial green algae, or marine invertebrate larvae (*Tamm, 1972*; *Chia et al., 1984*; *Brumley et al., 2015*).

A universal feature of multiciliated surfaces is the long-range beat synchronisation of individual cilia into metachronal waves (*Tamm, 1984*; *Brumley et al., 2012*; *Knight-Jones, 1954*; *Tamm, 1972*). Theoretical studies indicate that metachronal waves are energetically efficient and have a higher efficiency of flow generation than non-metachronal beating (*Osterman and Vilfan, 2011*; *Gueron and Levit-Gurevich, 1999*; *Elgeti and Gompper, 2013*). Metachronal coordination requires the orientation of ciliary beating planes during development (*Mitchell et al., 2007*; *Park et al., 2008*; *Mitchell et al., 2009*; *Guirao et al., 2010*; *Vladar et al., 2012*; *Kunimoto et al., 2012*). If cilia are oriented, synchronisation emerges by hydrodynamic coupling as adjacent cilia engage by flow (*Brumley et al., 2014*; *Elgeti and Gompper, 2013*). In addition, basal body coupling can also contribute to beat synchronisation as demonstrated in green algae (*Wan and Goldstein, 2016*).

However, biophysical mechanisms alone cannot explain all aspects of ciliary beat synchronisation. For example, the flow-networks generated by ependymal cilia show complex reorganisation that is

**eLife digest** The oceans contain a wide variety of microscopic organisms including bacteria, algae and animal larvae. Many of the microscopic animals that live in water use thousands of beating hair-like projections called cilia instead of muscles to swim around in the water. Understanding how these animals move will aid our understanding of how ocean processes, such as the daily migration of plankton to and from the surface of the water, are regulated.

The larvae of a ragworm called *Platynereis* use cilia to move around. Like other animals, *Platynereis* has a nervous system containing neurons that form networks to control the body. It is possible that the nervous system is involved in coordinating the activity of the cilia to allow the larvae to manoeuvre in the water, but it was not clear how this could work.

Here, Veraszto et al. investigated how *Platynereis* is able to swim. The experiments show that the larvae can coordinate their cilia so that they all stop beating at the same time and fold into to the body. Then the larvae can stimulate all of their cilia to resume beating. Veraszto et al. used a technique called electron microscopy to study how the nervous system connects to the cilia. This revealed that several giant neurons span the entire length of the larva and connect to cells that bear cilia. When these neurons were active, all the cilia in the body closed. When a different group of neurons in the larva was active, all of the cilia resumed beating. Together, these two groups of neurons were ultimately responsible for the swimming motions of the larvae.

Together, the findings of Veraszto et al. show that a few neurons in the nervous system of the larvae provide a sophisticated system for controlling how the larvae swim around. This suggests that the microscopic animals found in marine environments are a lot more sophisticated than previously appreciated. A next challenge is to find out how the neurons that control cilia connect to the rest of the animal's nervous system and how different cues influence when the larva swims or stops swimming. This would help us understand how the environment influences the distribution of animal larvae in the oceans and how this may change in the future.

under circadian regulation (*Faubel et al., 2016*). In mucociliary surfaces, changes in ciliary beat frequency are coordinated to regulate flow rates. Some of these changes are due to locally secreted diffusible molecules, including serotonin (*Maruyama et al., 1984*; *König et al., 2009*; *Walentek et al., 2014*) and neuropeptides (*Conductier et al., 2013*). However, long-range ciliary coordination is often under neuronal control (*Doran et al., 2004*; *Tamm, 2014*; *Arkett et al., 1987*; *Kuang and Goldberg, 2001*).

Ciliary swimmers display the most elaborate forms of neuronal ciliary coordination. In the ctenophore *Pleurobrachia*, prey-capture triggers a rapid synchronised beating of all eight ciliary combrows resulting in fast forward swimming (*Tamm, 2014*). In gastropod veliger larvae, the normal beating of velar cilia is periodically interrupted by coordinated, velum-wide ciliary arrests, triggered by calcium-spikes (*Arkett et al., 1987*). Similarly, annelid larvae show regular and coordinated arrests of the entire prototroch ciliary band (*Conzelmann et al., 2011*). Such coordinated changes in ciliary activity, often triggered throughout the whole body, require neuronal control.

The neuronal circuits coordinating whole-body ciliary activity have not been described in any animal. Here we reconstruct and functionally analyse the complete ciliomotor circuitry in the planktonic nectochaete larva of the marine annelid *Platynereis dumerilii*. *Platynereis* has emerged as a powerful model to study neural cell types and development (*Tomer et al., 2010*; *Marlow et al., 2014*; *Tessmar-Raible et al., 2007*) and the whole-body circuit bases of larval behaviour, including conical-scanning and visual phototaxis (*Jékely et al., 2008*; *Randel et al., 2014*). *Platynereis* nectochaete larvae (*Fischer et al., 2010*) have three trunk segments and use segmentally arranged ciliary bands to swim and muscles to turn while swimming (*Randel et al., 2014*) or to crawl on the substrate. Ciliary beating is also under neuromodulatory control and can be influenced by several neuropeptides and melatonin (*Conzelmann et al., 2011*; *Tosches et al., 2014*).

We used whole-body connectomics, transgenic neuron labelling, and calcium imaging to reconstruct and functionally analyse the entire ciliomotor system in *Platynereis* larvae. We identified a ciliomotor system consisting of interconnected catecholaminergic, cholinergic, and serotonergic

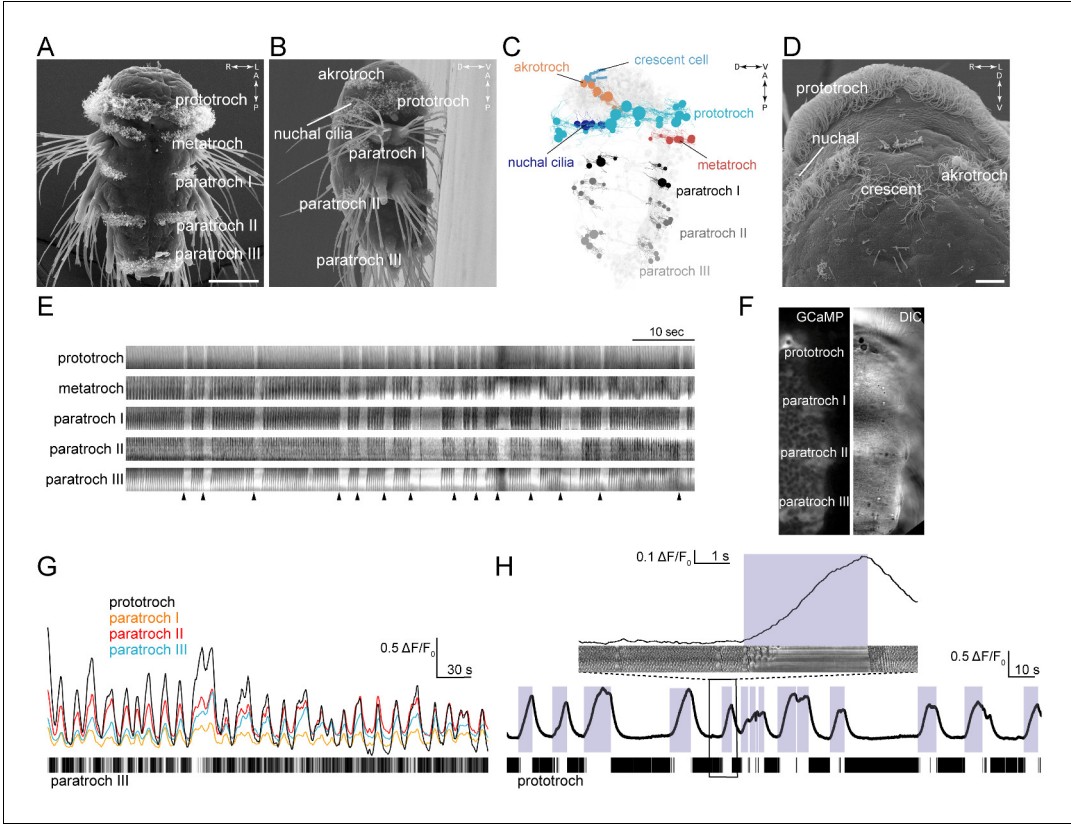

**Figure 1.** Coordinated ciliary beating and arrests in *Platynereis* larvae. (A–B,D) Scanning electron micrographs of 72 hr-post-fertilization larvae, (A) ventral view, (B) lateral view, (D) anterior view, close up. The structures dorsal to the crescent cell are non-motile sensory cilia. (C) Serial TEM reconstruction of ciliary band cells, lateral view. Nuclei of ciliary band cells are shown as spheres. The nervous system (brain and ventral nerve cord) is shown in light grey. (E) Kymographs of spontaneous ciliary activity from an immobilized 72 hr-post-fertilization larva. Arrowheads indicate the beginning of ciliary arrests. (F) Ventral view of the left half of a 72 hr-post-fertilization larva in a calcium-imaging experiment (left panel) with the corresponding differential interference contrast (DIC) image (right panel). (G) GCaMP6s signal from the prototroch and paratroch ciliary bands. A kymograph of ciliary activity in paratroch III is shown below. (H) GCaMP6s signal recorded from the prototroch of a 48 hr-post-fertilization larva at 45 frames per second. White areas in the kymograph correspond to periods of arrest. The boxed area is shown enlarged in the inset. Scale bars, 50 μm (A), 10 μm (D).

ciliomotor neurons. These neurons form a pacemaker system responsible for the whole-body coordination of alternating episodes of ciliary arrests and ciliary beating to regulate larval swimming.

## Results

### Coordinated beating and arrests of segmentally arranged locomotor cilia

*Platynereis* nectochaete larvae have segmentally arranged locomotor ciliated cells that form distinct ciliated fields (*Figure 1A–D*). The main ciliary band (prototroch) is located between the head and the trunk. The head has two patches of locomotor cilia (akrotroch) and multiciliated cells that cover the olfactory pit of the nuchal organs (nuchal cilia). There is also an unpaired multiciliated cell, the crescent cell, located anteriorly in the head in the middle of the sensory region of the apical organ. The three chaetigerous (with chaetae) trunk segments each have four fields of cilia forming the paratroch ciliary bands (paratroch I-III). There is an additional non-chaetigerous cryptic segment between the head and the trunk segments (*Steinmetz et al., 2011*) that harbours ventrally the metatroch ciliary band. Younger larvae have four posterior telotroch cells that merge with the paratroch III (eight

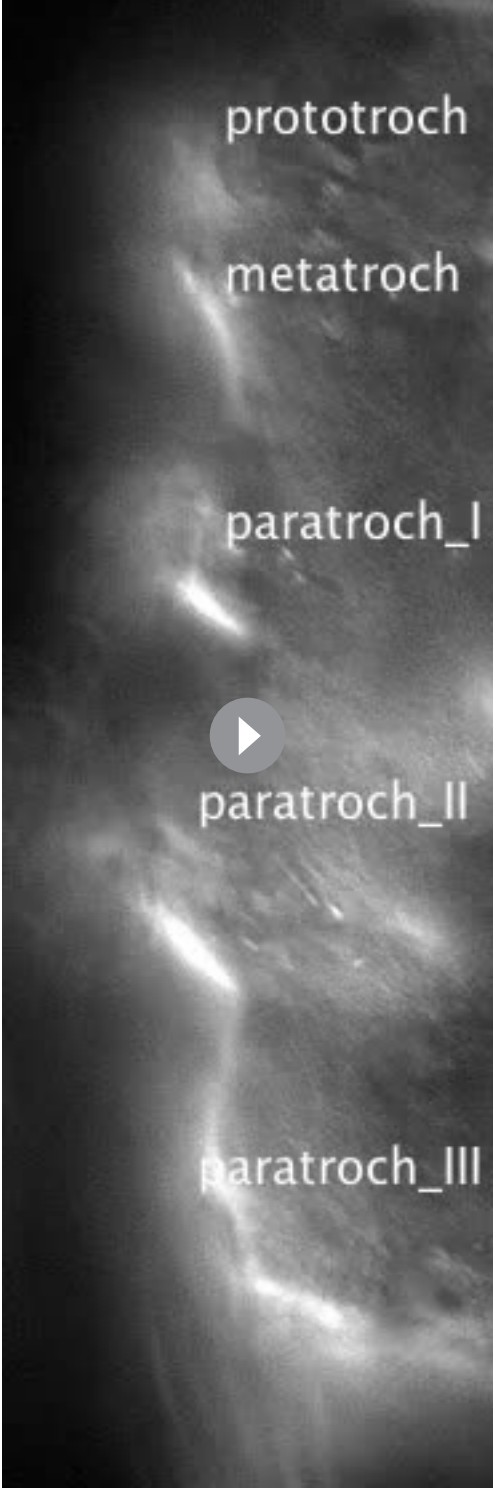

**Video 1.** Coordinated arrests of cilia on all ciliary band cells in a 72 hr-post-fertilization *Platynereis* larva. Differential interference contrast (DIC) imaging, recorded and played at 60 frames per second (fps).

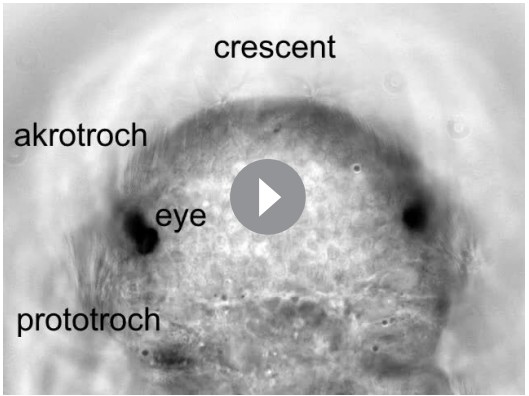

**Video 2.** Coordinated arrests of akrotroch and paratroch but not crescent cilia. DIC imaging of cilia in the *Platynereis* larval head. Recorded and played at 30 fps.

cells) during development to form a posterior ciliary band of 12 cells (*Starunov et al., 2015*) (referred to as paratroch III).

To investigate beat coordination across the ciliary bands in *Platynereis*, we imaged ciliary activity of mechanically immobilized 3-day-old nectochaete larvae. We found that spontaneous arrests of ciliary beating occurred synchronously across all ciliary bands (*Figure 1A–E* and *Video 1*). Cilia first resumed beating in the prototroch followed in synchrony by the paratroch cilia (with a delay of 19 ± 3 msec relative to the prototroch cilia). Arrests of prototroch cilia were also in synchrony with arrests in the akrotroch (*Video 2*) and the nuchal cilia (data not shown).

We next combined differential interference contrast (DIC) imaging with calcium imaging using a ubiquitously expressed fluorescent calcium indicator, GCaMP6s. In *Platynereis* larvae, regular arrests are triggered by bursts of spikes that can be recorded from the ciliary band cells (*Conzelmann et al., 2011*; *Tosches et al., 2014*). With calcium imaging, we detected periodic synchronous increases in calcium signals in cells of each ciliary band that corresponded with the arrests of cilia (*Figure 1G*, *Video 3*).

The only cilia that did not beat and arrest synchronously were the cilia of the crescent cell (*Figure 1D*). Although calcium signals also increase in the crescent cell simultaneously with the other multiciliated cells (data not shown), the cilia of this cell were beating when the other cilia were arrested (*Video 2*). Crescent cilia stopped beating after the locomotor cilia resumed beating (*Video 2*). The crescent cell has a lower

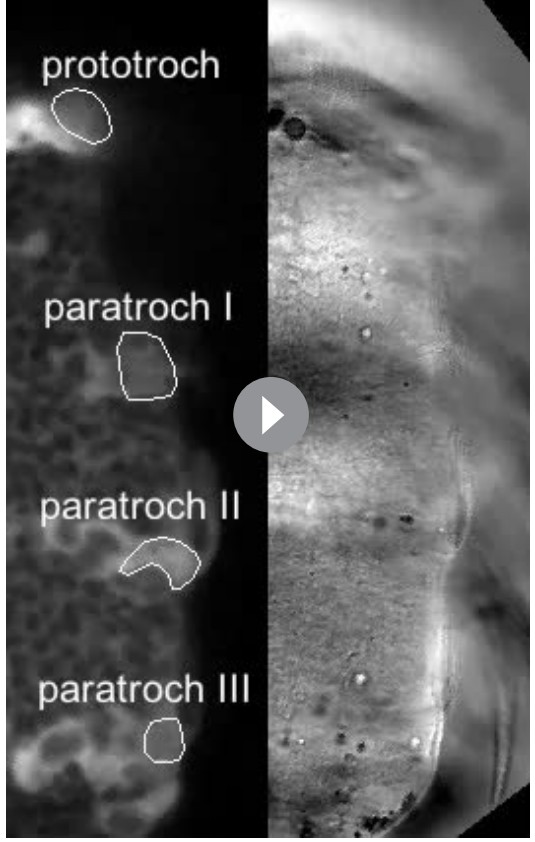

**Video 3.** Calcium and DIC imaging of ciliary bands in a 72 hr-post-fertilization *Platynereis* larva. The GCaMP6s signal (left panel) and DIC signal (right panel) are shown for the left side of the larva (ventral view). Regions of interest used to record calcium signals are indicated. Recorded at 3 fps, played at 25 fps.

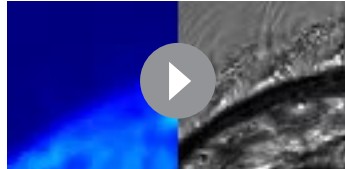

**Video 4.** Calcium and DIC imaging of the ciliary band in a 48 hr-post-fertilization *Platynereis* larva. The video was recorded at 45 fps, here only every 10th frame is shown at 45 fps.

density of cilia that beat more irregularly and do not organise into metachronal waves (*Video 2* and *Figure 1D*).

To investigate in more detail how intracellular calcium concentrations $[Ca^{2+}]_i$ in the prototroch cells relate to the arrests of cilia, we imaged calcium activity in close-ups from the prototroch cells at higher frame rates (*Video 4*). We found that ciliary activity did not correlate with the absolute levels of $[Ca^{2+}]_i$, but with the time derivative of calcium $d[Ca^{2+}]_i/dt$ (*Figure 1H*). As long as calcium levels increase, the cilia are arrested. As soon as calcium levels start to drop, the cilia resume their metachronal beating. A similar case was reported in sea urchin sperm where $d[Ca^{2+}]_i/dt$ rather than absolute $[Ca^{2+}]_i$ controls ciliary waveform during chemotaxis (*Alvarez et al., 2012*).

## Connectomic reconstruction of the ciliomotor circuitry

To understand the mechanism of ciliary beat synchronisation across ciliary bands and to get a comprehensive view of ciliomotor circuits in the *Platynereis* larva, we employed serial-section transmission electron microscopy (ssTEM). We used a whole-body ssTEM dataset of a 3 day old larva (*Randel et al., 2015*; *Shahidi et al., 2015*; *Shahidi and Jékely, 2017*) to identify and reconstruct all neurons innervating ciliary band cells.

First, we identified all multiciliated cells in this larva (*Figure 1C*). There are 23 cells in the prototroch that are arranged in an anterior and posterior tier, with one unpaired cell in the position of 11 o'clock (*Randel et al., 2014*). In the head, there are eight akrotroch cells, six ciliated cells in the nuchal organs, and an unpaired crescent cell. The metatroch consists of eight cells. In the first, second, and third trunk segments there are eight, 14, and 12 paratroch cells, respectively.

We next identified and reconstructed all neurons that were presynaptic to any of the multiciliated cells in the larva (*Jékely and Verasztó, 2017*). We define these neurons as ciliomotor neurons. Combining transgenic labelling, immunohistochemistry, and serial multiplex immunogold labelling (siGOLD) (*Shahidi et al., 2015*), we could assign a neurotransmitter and/or a neuropeptide to most of the ciliomotor neurons. We identified three main ciliomotor systems, consisting of either cholinergic, serotonergic, or mixed catecholaminergic/peptidergic neurons. We describe these in detail below.

## The cholinergic circuitry

We identified 11 cholinergic ciliomotor neurons in the larva (*Figure 2*, *Figure 2—figure supplement 1*). These include the previously described six ventral decussating cholinergic motorneurons (vMNs, including MN[r1], MN[r2], MN[r3], MN[l1], MN[l2], MN[l3]) and two rhabdomeric photoreceptor cells of the

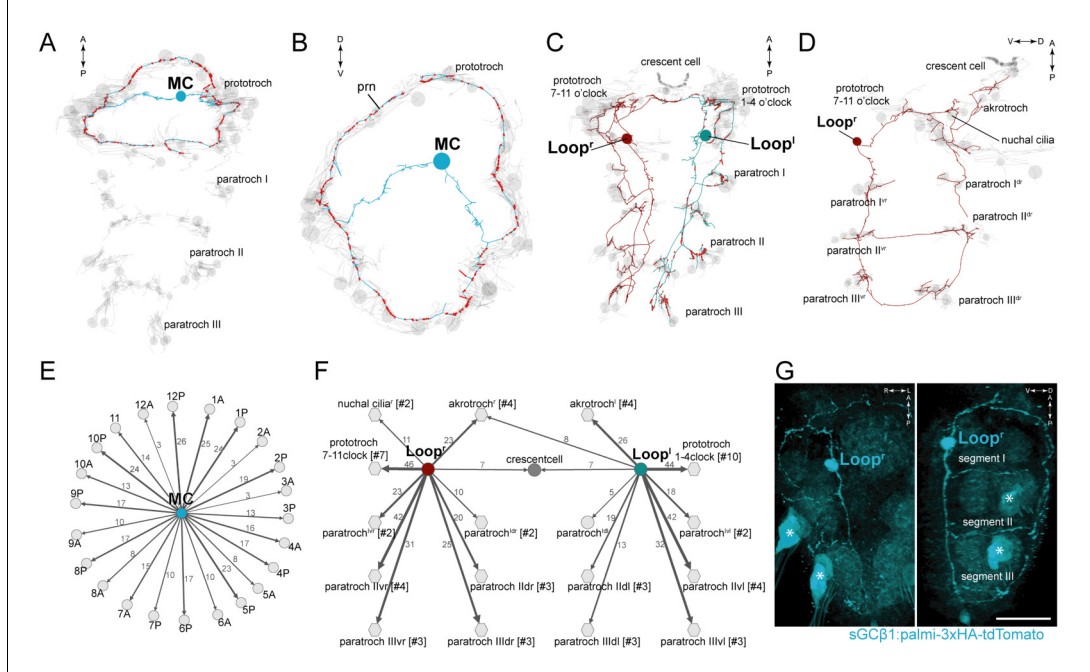

**Figure 2.** Anatomy and connectivity of biaxonal cholinergic ciliomotor neurons. (**A**) ssTEM reconstruction of the MC neuron (blue), ventral view. Ciliated cells are shown in grey. Circles represent position of cell body, lines represent axonal track. Presynaptic sites along the axon are marked in red. (**B**) The MC neuron in anterior view. (**C**) ssTEM reconstruction of the Loop neurons, ventral view. (**D**) The right Loop neuron in lateral view. (**E**) Synaptic connectivity of the MC neuron and the prototroch cells. (**F**) Synaptic connectivity of the Loop neurons and ciliary band cells. (**G**) Transgenic labelling of the right Loop neuron with an *sGCβ1:palmi-3xHA-tdTomato* construct. Expression was detected with anti-HA antibody staining. Right panel: ventral view, left panel: lateral view. Asterisks indicate background signal in the spinning glands. In the network graphs, nodes represent individual or grouped cells, as indicated by the numbers in square brackets. The edges show the direction of signalling from presynaptic to postsynaptic cells. Edges are weighted by the number of synapses. The number of synapses is indicated on the edges. Scale bar, 40 μm (**G**). Abbreviation: prn, prototroch ring nerve.

The following figure supplements are available for figure 2:

**Figure supplement 1.** Anatomy and connectivity of eyespotPRC[R3], ventral MN, and MN[ant] ciliomotor neurons.

**Figure supplement 2.** sGCbeta1 is a cholinergic marker.

larval eyespots (eyespotPRC[R3]) (*Figure 2—figure supplement 1*) (*Randel et al., 2014*, *2015*; *Jékely et al., 2008*). In addition to these neurons, we found three large biaxonal ciliomotor neurons, including a single ciliomotor neuron in the head, the MC neuron, and two trunk ciliomotor neurons that we call the Loop neurons (*Figure 2A–D*, *Video 5*).

The MC neuron has its soma in the centre of the larval head and projects two axons that run left and right towards the ciliary band. The axons branch at the prototroch and form dorsal and ventral branches that run along the prototroch ring nerve, a nerve running at the inside of the prototroch ring. The MC neuron is the most synapse-rich neuron in the entire *Platynereis* larval connectome (unpublished results) (*Randel et al., 2015*; *Shahidi et al., 2015*; *Randel et al., 2014*). We identified 341 presynaptic sites in the MC neuron. 335 of these target the ciliated cells of the prototroch ciliary band. The MC neuron is the only neuron in the body that is presynaptic to each of the 23 prototroch cells (3–25 synapses per prototroch cell) (*Figure 2B,E*, *Supplementary file 1*).

The somas of the Loop neurons are located in the first trunk segment. These neurons project one axon anteriorly and one posteriorly (*Figure 2C,D*, *Video 5*). The anterior axon joins the prototroch ring nerve and continues in the dorsal side of the larva where it runs posteriorly along a thin dorsal nerve. The posterior axon runs at the lateral edge of the ventral nerve cord, spans all three trunk segments and in the third segment runs to the dorsal side where it turns anteriorly. The anterior and

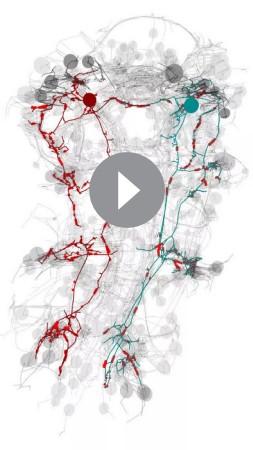

**Video 5.** The Loop ciliomotor neurons. ssTEM reconstruction of the Loop[l] and Loop[r] neurons. Ciliary bands and the ventral nerve cord are shown in grey. Synapses are marked in red.

posterior axons meet on the dorsal side. The Loop neurons thus loop around the entire body spanning the head and three segments in both the ventral and dorsal side. The Loop neurons have additional side branches in the head and in every segment. These project to and synapse on cells of the different ciliary bands. A single Loop neuron synapses on every ipsilateral paratroch cell, the akrotroch, the ciliated cells of the nuchal organ, and the lateral cells of the prototroch (*Figure 2F*). The Loop neurons also synapse on the crescent cell and represent its only synaptic input. Overall, the two Loop neurons innervate all locomotor ciliary fields in the larva, with the exception of the metatroch where we could not find any synaptic input at either side in the reconstructed specimen. In the prototroch, only some of the most dorsally and ventrally located prototroch cells (at 5, 6, and 12 o'clock position) lack synaptic input from the Loop neurons (*Figure 2F*, *Supplementary file 1*).

The anatomy and connectivity of the ventral cholinergic motorneurons (vMN) and the eyespot photoreceptors have already been described (*Jékely et al., 2008*; *Randel et al., 2014*, *2015*). We include them here for completion, and because we now have a more complete and revised reconstruction of the large intersegmental vMNs (*Figure 2—figure supplement 1*).

The vMNs form three left-right pairs (*Figure 2—figure supplement 1*). One pair (MN[r1] and MN[l1]) projects to the ventral nerve chord and innervates ventral paratroch cells and ventral longitudinal muscles. The second pair (MN[r2] and MN[l2]) projects dorsally and posteriorly along the trunk dorsal nerve and innervates lateral prototroch cells, dorsal paratroch cells, and dorsal longitudinal muscles. (Note that we swapped the name of MN[l1] and MN[l2] relative to (*Randel et al., 2015*) to reflect bilateral pairs). The third pair (MN[r3] and MN[l3]) is a purely ciliomotor pair (no muscle targets). Both cells synapse on lateral prototroch cells (only the posterior ring) and project to the ventral nerve cord.

The eyespotPRC[R3] cells synapse on two ipsilateral prototroch cells of the posterior prototroch belt (*Jékely et al., 2008*; *Randel et al., 2013*), representing two of the three direct sensory-ciliomotor neurons in the larva (*Figure 2—figure supplement 1*) (see below).

The cholinergic identity of the above neurons is supported by pharmacological experiments and by the expression of cholinergic markers (*Jékely et al., 2008*; *Randel et al., 2014*; *Tosches et al., 2014*; *Denes et al., 2007*) (*Figure 2—figure supplement 2*). To further test this at the single neuron level, we developed a transgenic reporter construct for cholinergic neurons. We could not obtain a working promoter construct for the canonical cholinergic markers *vesicular acetylcholine transporter* (*VAChT*) and *choline acetyltransferase* (*ChAT*), however, we identified a *soluble guanylyl cyclase-β* gene (*sGCβ1*) that coexpresses with *ChAT* as determined by in situ hybridisation combined with whole-body gene expression registration (*Asadulina et al., 2012*) (*Figure 2—figure supplement 2*). We cloned a 12 kB fragment directly upstream of the start site of *sGCβ1* and fused it with a tdTomato fluorescent reporter. The reporter was tagged with three tandem haemagglutinin (HA) tags and a palmitoylation sequence. We used zygote microinjection and mosaic transient transgenesis to label individual cholinergic neurons. In transgenic animals, we could label the Loop neuron (*Figure 2G*) and the ventral MN[l3] neuron (*Figure 2—figure supplement 2*), as identified by position and morphology, confirming their cholinergic identity. We could not label the MC neuron with this transgene, but its cholinergic identity is supported by *VAChT* expression in the cell body area of this cell (*Figure 2—figure supplement 2*) (*Jékely et al., 2008*) and also by pharmacological evidence (see below).

In addition to these 11 cholinergic cells, there are two biaxonal MN[ant] cells, described before (*Randel et al., 2015*), that are likely cholinergic. These cells have their cell bodies below the ciliary

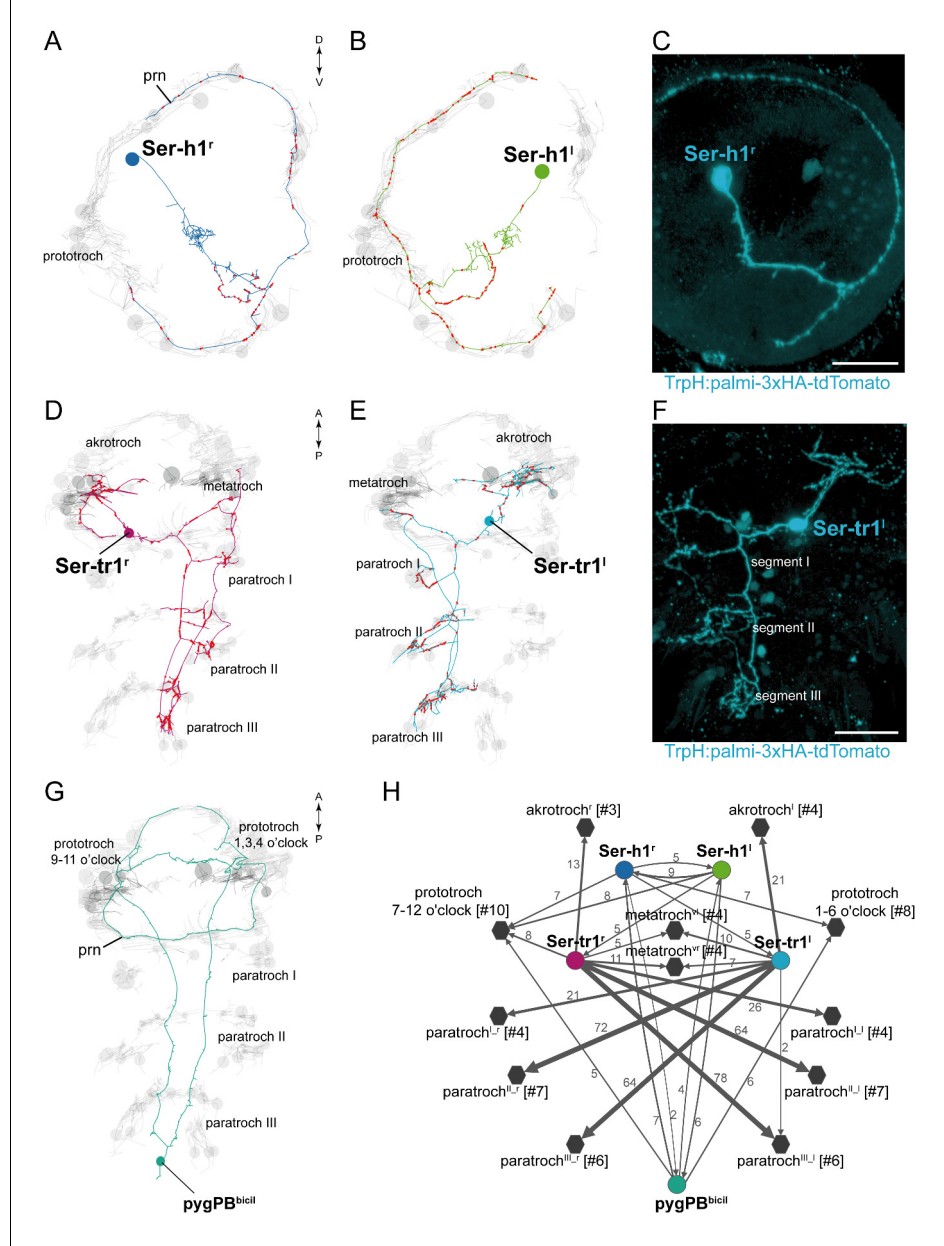

**Figure 3.** Anatomy and connectivity of serotonergic ciliomotor neurons. (**A–B**) ssTEM reconstruction of the right (**A**) and left (**B**) head serotonergic ciliomotor (Ser-h1) neuron, anterior view. (**C**) Transgenic labelling of the right Ser-h1 neuron with the *TrpH:palmi-3xHA-tdTomato* construct, anterior view. Expression was detected with anti-HA antibody staining. (**D–E**) ssTEM reconstruction of the right (**D**) and left (**E**) trunk serotonergic ciliomotor (Ser-tr1) cell, anterior view. (**F**) Transgenic labelling of the left Ser-tr1 neuron with the *TrpH:palmi-3xHA-tdTomato* construct, ventral view. (**G**) ssTEM reconstruction of the pygPB[bicil] serotonergic sensory-ciliomotor neuron. Ciliated cells are shown in grey. Circles represent position of cell body, lines represent axonal track. Presynaptic sites along the axon are marked in red. (**H**) Synaptic connectivity of the serotonergic ciliomotor neurons and the ciliary band cells. Nodes represent individual or grouped cells, as indicated by the numbers in square brackets. Edges show the direction of signalling from presynaptic to postsynaptic cells. Edges are weighted by the number of synapses. Synapse number is indicated. Edges with one synapse are not shown. Scale bars, 30 μm (**C, F**). Abbreviation: prn, prototroch ring nerve.

The following figure supplements are available for figure 3:

**Figure supplement 1.** Serotonergic neurons in *Platynereis* larvae.

*Figure 3 continued*

**Figure supplement 2.** Parallel axons of the Loop and Ser-tr1 neurons.

photoreceptor cells in a region of *VAChT* expression, but we could not verify their cholinergic identity at a single-cell resolution. The MN[ant] cells form synapses on the contralateral half of the prototroch, the metatroch, and the paratrochs (*Randel et al., 2015*) (*Figure 2—figure supplement 1*).

## The serotonergic circuitry

We identified five serotonergic ciliomotor neurons in the larva (*Figure 3*). There are two serotonergic ciliomotor neurons in the head (Ser-h1), with their somas lateral to the ciliary photoreceptor cells (*Figure 3A,B* and *Figure 3—figure supplement 1*). These neurons cross the midline and project along the contralateral side of the ciliary band, forming several synapses on the ciliated cells. The Ser-h1 cells also synapse reciprocally on each other in the brain neuropil.

In the trunk, there are two large serotonergic biaxonal neurons (Ser-tr1) with their somas in the first segment, near the somas of the cholinergic Loop neurons, anterior to the first commissure of the ventral nerve cord (*Figure 3D,E* and *Figure 3—figure supplement 1*). The Ser-tr1 and the Loop neurons represent the largest neurons in the entire *Platynereis* connectome (between 1,600–1,850 μm total cable length each). One of the axons of the Ser-tr1 cells projects anteriorly and innervates ipsilateral prototroch cells. The other axon crosses the midline in the first segment and its axon loops around the body, running in close proximity to the axons of the contralateral Loop neuron. The Ser-tr1 neurons also innervate every paratroch cell as well as the metatroch (*Figure 3H*, *Supplementary file 1*). The axons of the Ser-tr1 neurons run parallel to the axons of the Loop neurons throughout the body and are ultrastructurally distinct from the Loop neurons. The Ser-tr1 axons are characterised by a less electron-dense cytoplasm and the lack of dense core vesicles (abundant in the Loop neurons) (*Figure 3—figure supplement 2*).

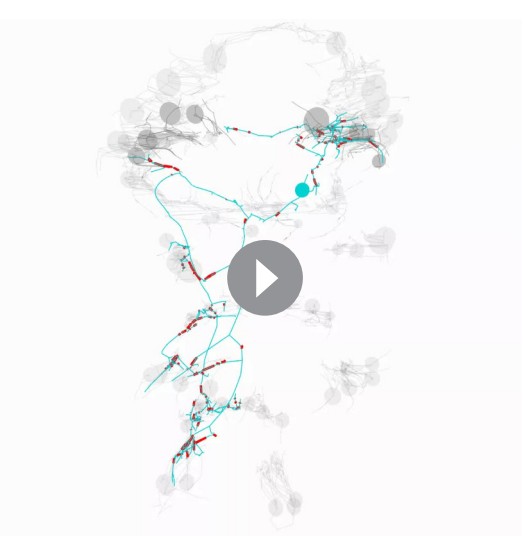

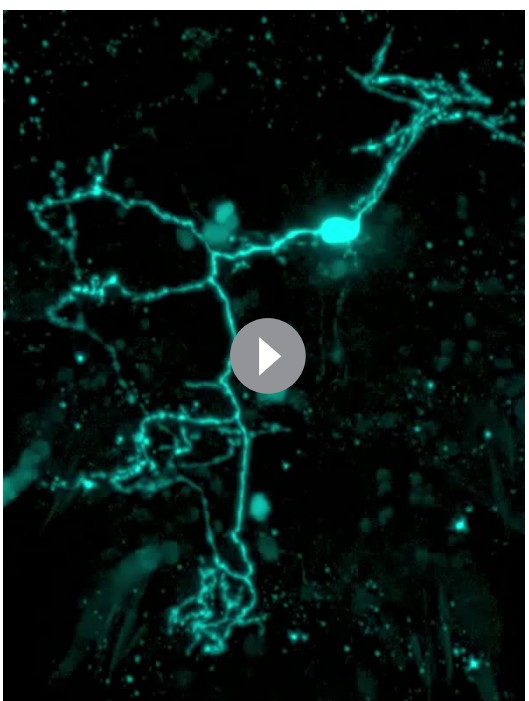

**Video 6.** The Ser-tr1 ciliomotor neurons. ssTEM reconstruction of the Ser-tr1[l] and Ser-tr1[r] neurons. Ciliary bands are shown in grey. The metatroch is shown in darker grey, synapses are marked in red.

**Video 7.** The Ser-tr1[l] neuron labelled by transient transgenesis. The *TrpH:palmy-3xHA-tdTomato* was used to label the Ser-tr1[l] neuron. The reporter was visualised by anti-HA antibody staining.

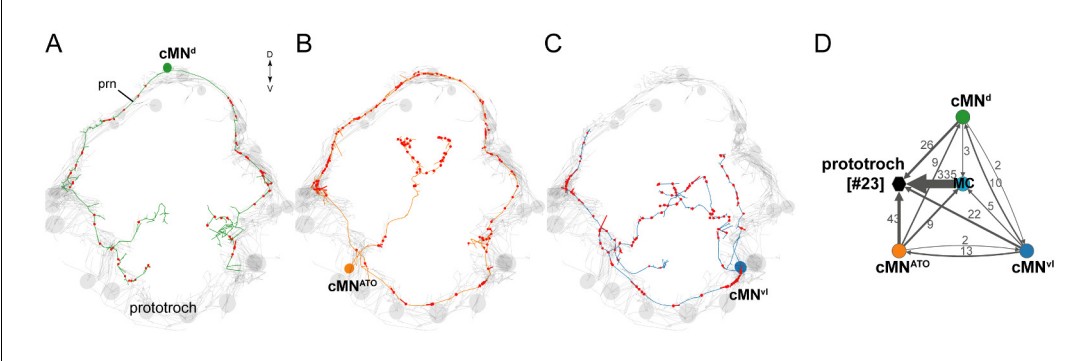

**Figure 4.** Anatomy and connectivity of catecholaminergic/peptidergic ciliomotor neurons. (**A–C**) ssTEM reconstruction of the cMN[d] (green) (**A**), cMN[ATO] (orange) (**B**), and cMN[vl] (blue) (**C**) ciliomotor neurons, anterior view. Ciliated cells are shown in grey. Circles represent position of cell bodies, lines represent axonal tracks. Presynaptic sites along the axon are marked in red. (**D**) Synaptic connectivity of the cMN neurons with the prototroch and the MC cell. Nodes represent individual or grouped cells, as indicated by the numbers in square brackets. Edges show the direction of signalling from presynaptic to postsynaptic cells. Edges are weighted by the number of synapses and indicate the direction of signalling. Synapse number is indicated. Abbreviation: prn, prototroch ring nerve.

The following figure supplement is available for figure 4:

**Figure supplement 1.** Catecholaminergic and neuropeptide marker expression in cMN neurons.

The Ser-tr1 cells were identified as serotonergic by comparing cell body positions and neuron morphologies to cells stained with a serotonin antibody in 2-day-old and 3-day-old larvae (*Figure 3— figure supplement 1*) (*Fischer et al., 2010*). We also used mosaic transient transgenesis with a construct containing a 5 kB upstream regulatory sequence of the *tryptophan hydroxylase* gene (*TrpH*), a marker of serotonergic neurons. We could label both the head Ser-h1 neurons and the Ser-tr1 cells (*Figure 3C,F*). The unique morphology of these cells and the correspondence of the TEM reconstruction (*Video 6*, *Video 7*) to the transgenic labelling confirm the serotonergic identity of these cells.

We also identified a giant serotonergic sensory-motor neuron (pygPB[bicil]) with a cell body in the pygidium and a unique morphology (*Figure 3G*). This cell has two sensory cilia and two axons that run anterior along the ventral nerve chord. The axons then innervate the prototroch cilia in the head and send a branch to the anterior nervous system. The pygPB[bicil] neuron is identifiable by serotonin immunolabeling (*Figure 3—figure supplement 1*) and also expresses the neuropeptide FMRFamide as determined by serial multiplex immunogold (siGOLD) labelling (*Shahidi et al., 2015*). This cell, together with the eyespot photoreceptor cells, is the third sensory neuron that directly synapses on ciliated cells.

## Catecholaminergic and peptidergic ciliomotor neurons

There are three molecularly distinct, asymmetric peptidergic cells (cMN[vl], cMN[d], cMN[ATO]) with their cell bodies at the level of the prototroch ciliary band (*Figure 4*). (cMN[vl] was referred to in [*Shahidi et al., 2015*] as cMN[PDF-vcl1]). The two ventral-lateral cells (cMN[vl], cMN[ATO]) express *tyrosine hydroxylase*, a marker of dopaminergic neurons (*Figure 4—figure supplement 1*). The cMN[ATO] cell also expresses *dopamine-β-hydroxylase*, indicating that it is noradrenergic (*Figure 4—figure supplement 1*).

The cMN cells express different combinations of neuropeptides. The neurons cMN[vl] and cMN[d] express pigment dispersing factor (PDF), as determined by siGOLD (*Shahidi et al., 2015*). The cMN[ATO] cell expresses allatotropin and its full morphology is revealed by immunostaining with an allatotropin antibody (*Figure 4—figure supplement 1*). The cMN[vl] neuron also expresses FLamide and FVamide, cMN[ATO] expresses DLamide, and cMN[d] expresses the neuropeptides L11, and LYamide, as revealed by immunostainings (*Conzelmann et al., 2011*) and in situ hybridisation (*Figure 4—figure supplement 1*). The cMN cells synapse on prototroch cells and the MC cell and also synapse reciprocally among themselves (*Figure 4D*, *Supplementary file 1*).

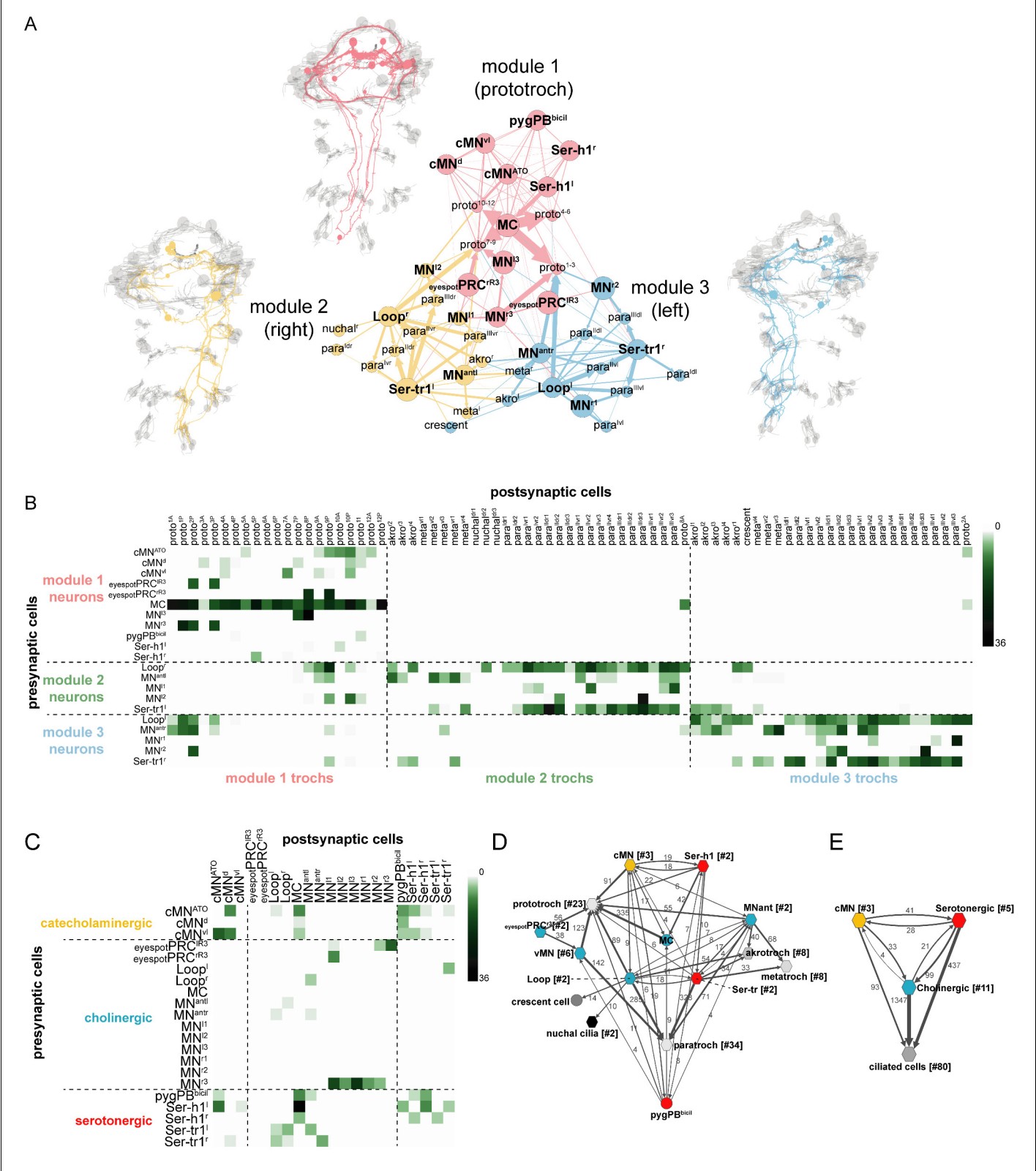

**Figure 5.** Overall connectivity of ciliomotor neurons and ciliary bands. (**A**) Synaptic connectivity graph of all ciliomotor neurons and ciliary band cells. The network is partitioned into three modules. The neurons belonging to each module are shown in the anatomical reconstructions. The ciliary band cells are grouped into anatomical units. (**B**) Matrix representation of the connectivity of ciliomotor neurons to all ciliated cells. Colour intensity is proportional to the number of synapses. (**C**) Matrix representation of the connectivity of ciliomotor neurons amongst themselves. Colour intensity is

*Figure 5 continued on next page*

*Figure 5 continued*

proportional to the number of synapses. (D–E) Graph representations of the ciliomotor circuit. Nodes with <4 synapses are not shown. In (D) nodes are grouped by cell type (teal, cholinergic; red, serotonergic; orange, cMN; grey, ciliated cells). In (E) all cMN, serotonergic, cholinergic neurons, and ciliary band cells are represented as one node each. In (D, E) edges with <4 synapses are not shown. Abbreviations: proto, prototroch; meta, metatroch; para, paratroch; akro, akrotroch.

## Overall connectivity of the ciliomotor system

To analyse the overall connectivity of the ciliomotor system (*Video 8*, *Supplementary file 1*), we next performed modularity analysis to identify more strongly connected communities of cells (*Blondel et al., 2008*). We could subdivide the network of ciliomotor neurons and multiciliated cells into three communities each containing cells that were more strongly connected among each other (*Figure 5A*, *Supplementary file 1*). The three modules can be distinguished primarily based on the ciliated cells they contain (*Figure 5B* and *Supplementary file 1*). Module 1 includes ciliomotor neurons that only innervate prototroch cells (MC, cMN, Ser-h1, and pygPB[bicil], MN[l3], MN[r3], eye-spotPRC[R3], and most prototroch cells). Module 2 is composed of the right (ipsilaterally-projecting) Loop[r] neuron and the left (contralaterally projecting) Ser-tr1[l], MN[antl], MN[l1], and MN[l2] neurons. These neurons share ciliated target cells that are primarily on the right body side (*Figure 5A,B*). Module 3 includes neurons providing innervation primarily to cilia on the left body side (Loop[l], Ser-tr1[r], MN[antr], MN[r1], and MN[r2]). Module 2 and 3 thus contain the corresponding set of left-right neuron pairs and their ciliary targets.

There are also many interconnections between ciliomotor neurons expressing distinct transmitters (cholinergic, serotonergic, and catecholaminergic/peptidergic) (*Figure 5C–E*, *Supplementary file 1*). The cMN neurons are presynaptic to the MC neuron, the MN[ant] and the Loop neurons, and show reciprocal connectivity with the Ser-h1 and pygPB[bicil] cells. The serotonergic and cholinergic systems are also interconnected. The Ser-h1 cells are presynaptic to the MC neuron and the Ser-tr1 and Loop neurons are reciprocally connected. The pygPB[bicil] neuron is also presynaptic to the MC cell (*Figure 5C–E*).

## Rhythmic activation of cMN neurons in and out of phase with ciliary arrests

To functionally characterise the ciliomotor system, we performed calcium-imaging experiments in immobilized larvae ubiquitously expressing GCaMP6s. Since 3-day-old larvae are difficult to immobilize without compromising neuronal activity, we used 2-day-old larvae for activity imaging. At this stage, the musculature is not yet well developed. However, several ciliomotor neurons are already present as evidenced by immunostaining or in situ hybridisation (cMN, serotonergic, cholinergic cells) (*Figure 3—figure supplement 1*, *Figure 4—figure supplement 1*).

Calcium imaging revealed that the cMN neurons display spontaneous rhythmic activity patterns. The somas of the cMN cells are close to the prototroch cells at unique positions allowing their unambiguous identification. We found that the cMN[vl] and cMN[d] neurons showed rhythmic increases in the GCaMP signal in synchrony with the ciliary band (*Figure 6*). In contrast, the activity of the cMN[ATO] was negatively correlated with the two other cMN neurons and the ciliary band (*Figure 6C,D,F*, *Figure 6—source data 1*). We could detect the rhythmic activation of the cMN

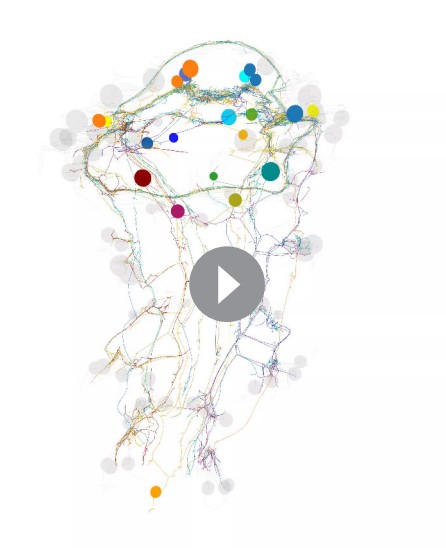

**Video 8.** The entire ciliomotor circuit of the *Platynereis* larva. Neuron reconstructions of all ciliomotor neurons.

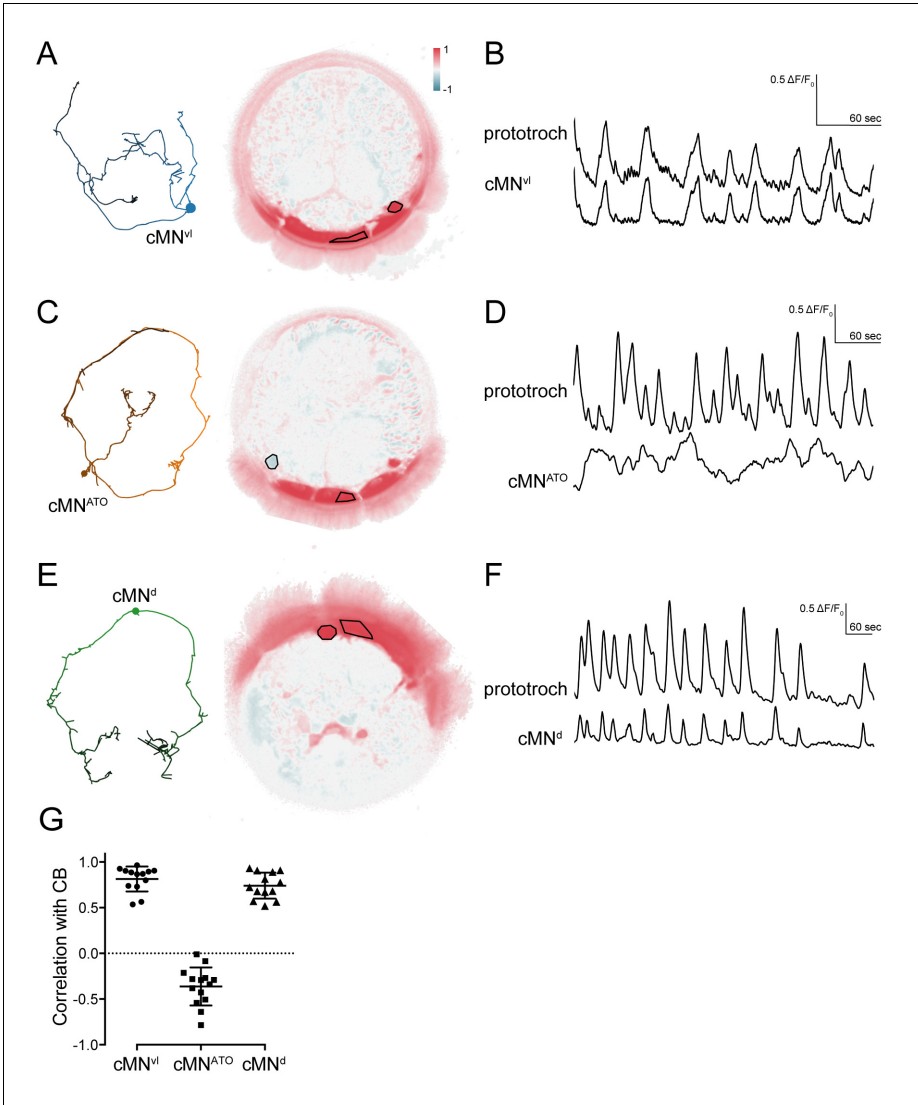

**Figure 6.** Activity of cMN neurons. (**A**) ssTEM reconstruction (left) and correlation (Pearson's r) map of neuronal activity (right) of cMN$^{vl}$, anterior view. Correlation values were calculated relative to the prototroch (outlined). (**B**) Calcium-imaging trace of cMN$^{vl}$ and the prototroch (**C**) ssTEM reconstruction (left) and correlation (Pearson's r) map of neuronal activity (right) of cMN$^{ATO}$. Correlation values were calculated relative to the prototroch (outlined). (**D**) Calcium imaging trace of cMN$^{ATO}$ and the prototroch. (**E**) ssTEM reconstruction (left) and correlation (Pearson's r) map of neuronal activity (right) of cMN$^{d}$. Correlation values were calculated relative to the prototroch (outlined). (**F**) Calcium-imaging trace of cMN$^{d}$ and the prototroch. (**G**) Correlation of GCaMP signals of cMN neurons with GCaMP signals measured from the prototroch ciliary band (CB). Data points represent measurements from different larvae, n > 12 larvae. Mean and standard deviation are shown. All sample medians are different from 0 with p-values≤0.0002 as determined by Wilcoxon Signed Rank Test.

The following source data is available for figure 6:

**Source data 1.** Source data for *Figure 6G* with correlation values of neuronal activity.

neurons in larvae from 24 hr-post-fertilisation (hpf) onwards.

## The MC and Loop neurons trigger coordinated ciliary arrests

Next, we imaged the activity of the cholinergic MC and Loop neurons. We could identify the MC neuron based on its position and biaxonal morphology as revealed by the GCaMP signal (*Video 9*).

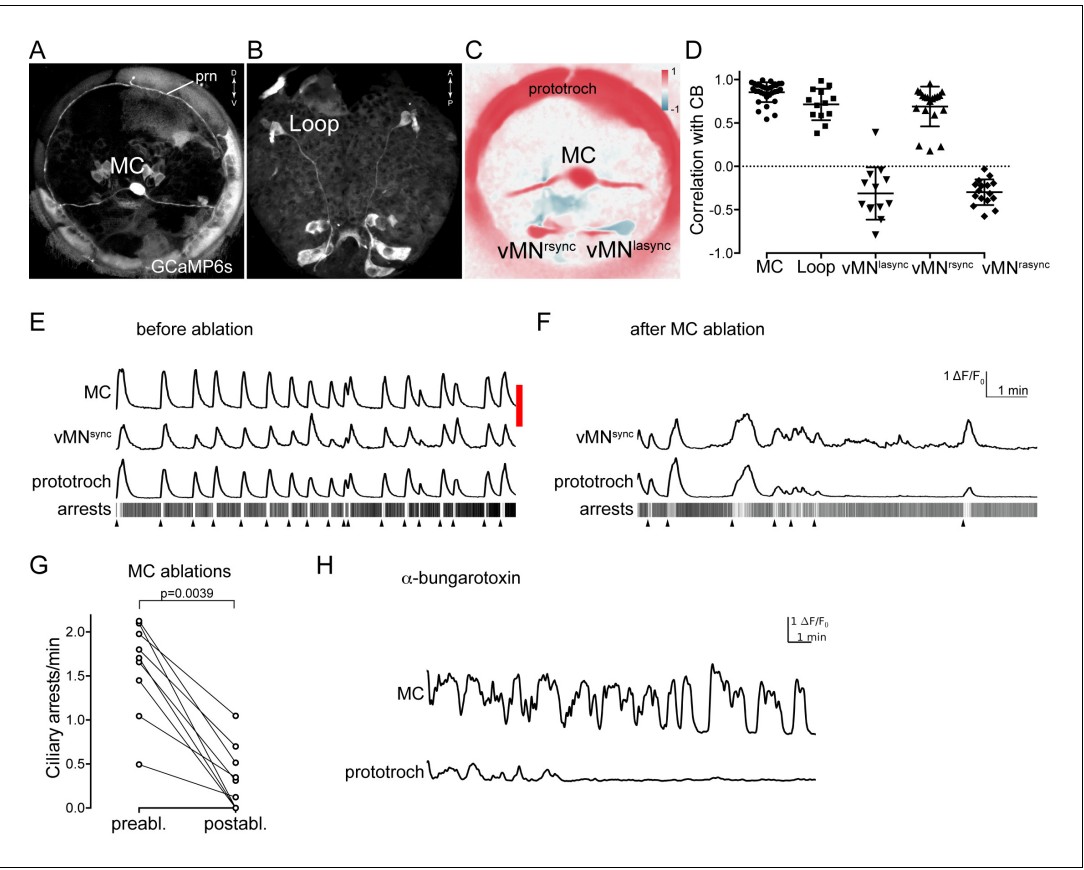

**Figure 7.** Activity of cholinergic neurons. (**A–B**) GCaMP6s signal revealing cell morphologies of the active MC (anterior view) (**A**) and Loop (ventral/posterior view) (**B**) neurons in 2 days-post-fertilisation larvae. (**C**) Correlation (Pearson's r) map of neuronal activity of the MC, vMN$^{sync}$ and vMN$^{async}$ neurons. Correlation values were calculated relative to the prototroch. (**D**) Correlation of GCaMP signals of cholinergic neurons with GCaMP signals measured from the prototroch ciliary band (CB). Data points represent measurements from different larvae, n > 12 larvae. Mean and standard deviation are shown. All sample medians are different from 0 with p-values≤0.0034 as determined by Wilcoxon Signed Rank Test. (**E–F**) Effect of MC neuron ablation on prototroch activity and ciliary closures. Red bar represents the time of the ablation. GCaMP signals in the MC neuron, in a vMN$^{sync}$ neuron, and in the prototroch before (**E**) and after (**F**) MC neuron ablation. (**G**) Number of ciliary arrests in the prototroch before and after MC neuron ablation. (**H**) Calcium signals measured from the MC neuron and the prototroch following the addition of 50 µM alpha-bungarotoxin. Abbreviation: prn, prototroch ring nerve.

The following source data and figure supplements are available for figure 7:

**Source data 1.** Source data for *Figure 7D* with correlation values of neuronal activity.

**Source data 2.** Source data for *Figure 7G* with ciliary closures before and after MC neuron ablation.

**Figure supplement 1.** Characterisation of MC neuron activity.

**Figure 7–figure supplement source data 1.** Source data for *Figure 7—figure supplement 1A* with calcium imaging data of the MC neuron from 96 individual larvae.

**Figure supplement 2.** Effect of MC neuron ablation on prototroch activity and ciliary closures.

We found that the activity of the MC neuron was periodic and strongly correlated with the activity of the prototroch cells and with ciliary arrests (*Figure 7A,D*, *Figure 7—source data 1*). Recording from 96 different larvae (*Figure 7—figure supplement 1*, *Figure 7—figure supplement 1—source data*

 

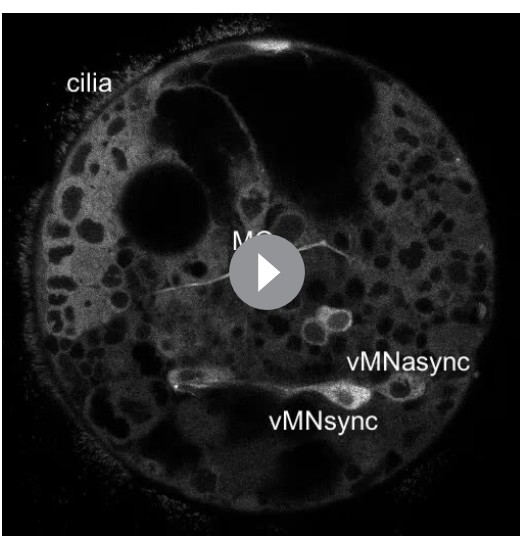

**Video 9.** Calcium imaging of the MC neuron in a 30 hr-post-fertilisation larva. Ubiquitously expressed GCaMP6s was used to image neuronal activity. The MC neuron, a vMN<sup>sync</sup>, a vMN<sup>async</sup> neuron, and prototroch cilia are labelled. The video was recorded at 1.5 fps and is shown at 45 fps.

1) followed by Fourier analysis of the calcium signals revealed a median cycle length of 71 s of the periodic activation of the MC neuron. In 43% of larvae the MC neuron had a cycle length between 50–100 s (*Figure 7—figure supplement 1*).

In the trunk, we also identified two cells in the first segment that were activated synchronously with the ciliary band cells and thus the MC neuron (though not imaged simultaneously for technical reasons). These two trunk cells had somas in the approximate position of the two Loop neurons with ipsilaterally projecting axons, and could thus be identified as the Loop neurons (*Figure 7B,D*, see also *Figure 8—figure supplement 1A*).

In the connectome graph, the Loop and the MC neurons provide the main cholinergic motor input to the ciliary bands. Their coordinated activation and high number of synapses suggest that these cells are responsible for triggering the coordinated arrests of locomotor cilia across the body. To test this experimentally, we focused on the MC cell. First, we ablated the MC cell with a pulsed laser in 24–36 hra-post-fertilisation larvae. MC-cell ablation eliminated the rhythmic activity in the prototroch cells and thus most ciliary arrests (*Figure 7E,F* and *Figure 7—figure supplement 2*, *Figure 7—source data 2*). This shows that the MC cell is essential for triggering the rhythmic arrests of the prototroch cilia. To test whether acetylcholine is involved in rhythm generation and triggering ciliary arrests, we incubated larvae in alpha-bungarotoxin, a specific blocker of nicotinic acetylcholine receptors. When we imaged alpha-bungarotoxin-treated larvae, we observed the decoupling of the activity of the MC neuron and the ciliary band. While the MC neuron continued to rhythmically activate, the calcium signals were progressively reduced and then disappeared in the ciliary band (*Figure 7H*). Ciliary arrests were also eliminated (data not shown). This indicates that cholinergic signalling is not required for the generation of the neuronal rhythm but is required for ciliary arrests.

The ventral cholinergic motorneurons (vMN) were previously shown to rhythmically activate, and were suggested to trigger ciliary arrests (*Tosches et al., 2014*). To directly correlate the activity of these cells to ciliary activity we imaged from the ventral MN cells and the prototroch cells simultaneously. We could identify three ventral MN cells that were activated either synchronously or asynchronously (designated vMN<sup>sync</sup> and vMN<sup>async</sup>) with the ciliary band (*Figure 7C–E*). We cannot unambiguously link these three cells to the six ventral MN cells identified by EM, but based on cell body positions and their distinct cell lineage as identified by blastomere injections (*Tosches et al., 2014*), the ventral-most cells likely correspond to MN<sup>r3</sup> and MN<sup>l3</sup>. Following MC neuron ablation, the activity of the prototroch correlated with the activity of the vMN<sup>sync</sup> neuron suggesting that this cholinergic neuron can maintain a lower rate of prototroch activation.

## Serotonergic neuron activity is anti-correlated with cholinergic neuron activity

The cholinergic and serotonergic ciliomotor neurons have many synaptic connections, suggesting that their activity is interdependent (*Figure 5*). Correlation analysis of the calcium-imaging videos revealed that the head serotonergic cells (Ser-h1), identified by position and morphology, showed anti-correlated activity with the ciliated cells and the MC neuron (*Figure 8A,B,E*, *Figure 8—source data 1*).

In the ventral nerve cord, we also identified two neurons whose activity was anti-correlated with the activity of the ciliary band. (*Figure 8C–E*). The GCaMP signal did not reveal the axonal

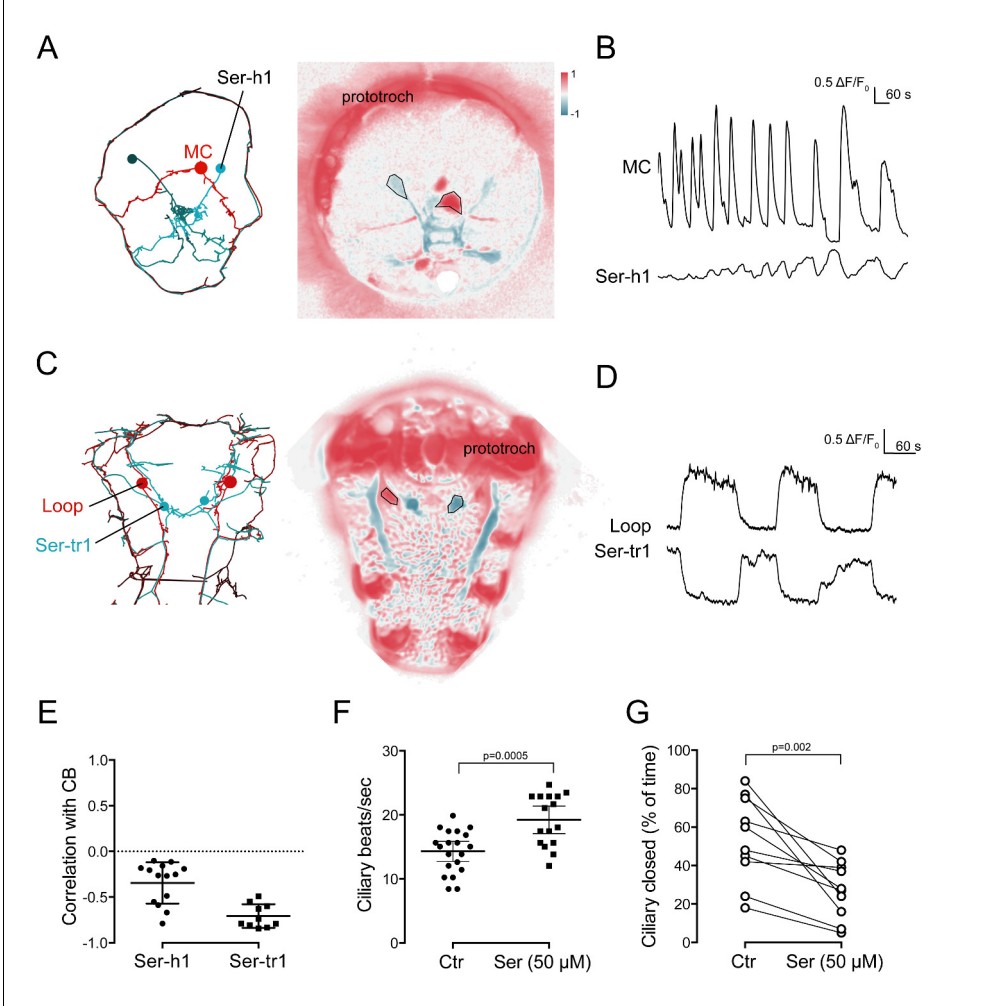

**Figure 8.** Activity of serotonergic neurons. (**A**) ssTEM reconstruction (left) and correlation (Pearson's r) map of neuronal activity (right) of the Ser-h1 neurons, anterior view. Correlation values were calculated relative to the prototroch. (**B**) Calcium signals measured from a Ser-h1 neuron and the MC cell. (**C**) ssTEM reconstruction (left) and neuronal activity correlation (right) of the Ser-tr1 neurons, ventral view. Correlation values were calculated relative to the prototroch. (**D**) Calcium signals measured from a Ser-tr1 neuron and a Loop neuron. (**E**) Correlation of GCaMP signals of serotonergic neurons with GCaMP signals measured from the prototroch ciliary band (CB). Data points represent measurements from different larvae, n > 9. Mean and standard deviation are shown. All sample medians are different from 0 with p-values≤0.002 as determined by Wilcoxon Signed Rank Test. (**F**) Ciliary beat frequency of prototroch cilia in the absence and presence of serotonin. (**G**) Ciliary closures of prototroch cilia in the absence and presence of serotonin. P-values of an unpaired (**F**) and paired (**G**) t-test are shown. In (**F**,**G**) n > 9 larvae. Samples in (**F**,**G**) passed the D'Agostino and Pearson omnibus normality test (alpha = 0.05).

The following source data and figure supplement are available for figure 8:

**Source data 1.** Source data for *Figure 8E* with correlation values of neuronal activity.
**Source data 2.** Source data for *Figure 8F*.
**Source data 3.** Source data for *Figure 8G*.
**Figure supplement 1.** In vivo labelling of a Ser-tr1 cell by photoactivation during calcium imaging.

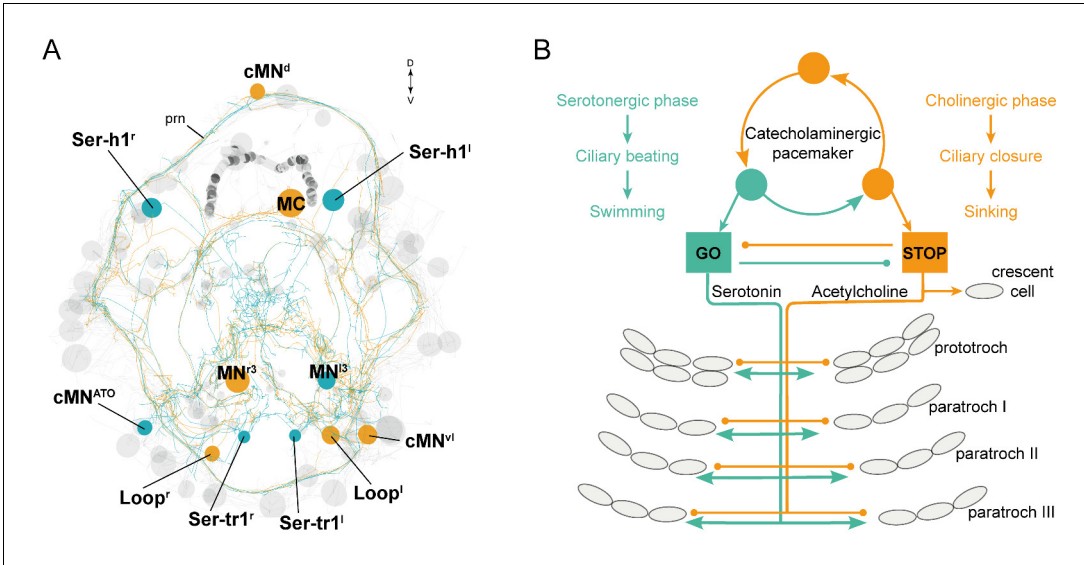

**Figure 9.** The stop-and-go ciliomotor system of the *Platynereis* larva. (**A**) ssTEM reconstruction of ciliomotor neurons that are activated synchronously (orange) and asynchronously (teal) with the ciliary bands. (**B**) Schematic of the circuitry, consisting of a pacemaker system, as well as cholinergic and serotonergic ciliomotor neurons that span the entire body and innervate all multiciliated cells. Abbreviation: prn, prototroch ring nerve.

morphology of these neurons, but their cell-body position corresponded to the position of the Ser-tr1 cells. To directly test the identity of these cells, we performed in vivo photoactivation experiments. We co-injected GCaMP mRNA with an mRNA encoding a photoactivatable mCherry fluorescent protein (PAmCherry1). During calcium imaging, we identified the Ser-tr1 candidate neuron based on its activity and then photoactivated PAmCherry1 in this neuron. The larvae were subsequently fixed and stained for serotonin. In all cases (n = 11 larvae) we labelled the serotonergic Ser-tr1 neuron, together with a few out-of-focus cells that were labelled more weakly (*Figure 8—figure supplement 1*).

To further explore how serotonin influences ciliary activity, we treated larvae with serotonin. Serotonin increased ciliary beat frequency and inhibited ciliary arrests compared to untreated larvae (*Figure 8F,G*, *Figure 8—source data 2*, *Figure 8—source data 3*).

## Discussion

The three-segmented *Platynereis* larva has many distinct cell groups with locomotor cilia that efficiently propel the larva in water. Periods of swimming are interrupted by ciliary arrests. Here we investigated how spontaneous ciliary arrests and beating are coordinated across ciliated fields.

With whole-body connectomics and calcium imaging, we described the anatomy, connectivity, and activity of the entire ciliomotor system in the *Platynereis* larva. In trochophore larvae, which do not yet have a functioning musculature, this system represents the entire motor part of the nervous system.

One of the most fascinating aspects of this ciliomotor system is its rhythmic pattern of activation consisting of two alternating phases. In one phase, a dopaminergic, a peptidergic, and several cholinergic neurons are active together with the ciliary band cells. In the other phase, anti-correlated with the cholinergic rhythm, one noradrenergic neuron and the serotonergic neurons are active. Our connectome and imaging data indicate the rhythm is generated by the interactions of the ciliomotor neurons themselves. This is supported by the presence of many and often reciprocal synaptic connections between the ciliomotor motorneurons. We cannot exclude the possibility that interneurons connecting distinct ciliomotor neurons are involved in the generation of the rhythmic pattern and the synchronisation of ciliomotor activity. We could identify such interneurons in the connectome

(unpublished results). However, we could not observe rhythmic activity for any of these interneurons by calcium imaging.

We could subdivide the ciliomotor circuit into three connectivity modules. All three modules contain both serotonergic and cholinergic neurons and have largely non-overlapping ciliary targets (prototroch, left and right ciliated fields, respectively). Module 1 is special in that besides the serotonergic and cholinergic neurons it also contains the three unpaired peptidergic cMN neurons, two of which are also catecholaminergic. These neurons are likely involved in the generation of the rhythmic pattern. Three observations support this. First, together with the MC neuron, the cMN cells are the first neurons that show a rhythmic pattern during development. Second, they contain three cells with distinct transmitter profiles suggesting the possibility of antagonistic reciprocal signalling. Third, they are presynaptic or reciprocally connected with all the other ciliomotor neurons. For example, the cMN neurons are presynaptic to both the MC and the Loop neurons that are themselves only weakly connected, yet activate synchronously. We can also exclude a role for cholinergic neurons in rhythm generation since blocking cholinergic transmission did not eliminate the rhythmic activation of the MC neuron. The precise delineation of the pacemaker neurons will require further cell ablation studies.

The function of the rhythmically active ciliomotor circuitry is to trigger the coordinated arrests of all cilia in the body and likely to trigger the coordinated resumption of beating. Targeted neuron ablation revealed a critical role for the cholinergic MC neuron in coordinated arrests of prototroch cilia suggesting that this neuron relays the rhythmic activation of the cMN neurons to trigger arrests. Our imaging and connectome data suggest that the MC neuron together with the two Loop neurons is able to trigger coordinated, body-wide ciliary arrests. These three cells innervate all multiciliated cells in the body (with the exception of the metatroch). Arrests of metatroch cilia may be triggered by the MN$^{ant}$ neurons, the only putatively cholinergic neurons that innervate the metatroch. The metatroch is a special ciliary band that in some feeding trochophore larvae beats in opposition to the prototroch to collect food particles (*Rouse, 1999*; *Pernet and Strathmann, 2011*). The distinct pattern of innervation for the metatroch may reflect the unique evolutionary history of this ciliary band.

The crescent cell represents another specialised multiciliated cell. This cell shows a beat pattern alternating with the beating of locomotor cilia. Crescent cilia thus beat when the larvae do not swim. The only synaptic contacts to the crescent cell come from the Loop neurons. The crescent cell also shows increases in calcium signals, in synchrony with the ciliary bands. However, crescent cilia respond differently to stimulation than locomotor cilia and start beating instead of arresting. The crescent cell is in the middle of the apical organ, a sensory area. Beating of the crescent cilia may serve to generate water flow to facilitate chemosensory sampling in the apical organ, similar to the role of cilia-generated flows in olfaction in fish (*Reiten et al., 2017*).

Parallel and upstream to the cholinergic system, we identified four serotonergic ciliomotor neurons that also collectively innervate all ciliary bands. Serotonergic cells are also presynaptic to the cholinergic cells and serotonin application suppresses closures and increases ciliary beat frequency. Determining the site of serotonin action would require further studies. One possibility is that serotonin influences the pacemaker activity thereby indirectly affecting closures. Serotonin may also act directly on ciliated cells via serotonergic synapses to increase beat frequency. The activation of the serotonergic cells corresponds to periods of swimming and thus these cells are likely responsible for the resumption of beating.

The monoamines and neuropeptides we mapped to the ciliomotor neurons may act at synapses or extrasynaptically (*Bentley et al., 2016*). Understanding the details of signalling will require the identification of the receptor expressing cells. We recently characterised several neuropeptide receptors, adrenergic receptors and other monoamine receptors in *Platynereis* (*Bauknecht and Jékely, 2017*, *2015*). We also developed a method to dissect chemical connectivity of neuropeptide signalling with cellular resolution (*Williams et al., 2017*). These approaches will help to dissect the mechanisms of rhythm generation and the modulation of the rhythm and ciliary activity.

What is the relevance of coordinated ciliary arrests for larval behaviour? It was shown that in freely-swimming *Platynereis* larvae the alternation of periods of upward swimming (when cilia beat) and sinking (when cilia are arrested) is important for maintaining a constant depth in the water column of the negatively buoyant larvae (*Conzelmann et al., 2011*). It is remarkable that the *Platynereis* larval nervous system exhibits a rhythmic autonomic activity already in the 1-day-old stage, when the

nervous system only consists of <20 differentiated neurons. This ongoing activity is present in all larvae and is not induced by sensory input. Early larval behaviour is thus best described in operant terms (*Brembs, 2017*). This autonomic activity can then be modified by sensory input. Several neuromodulators, including neuropeptides and melatonin were shown to either inhibit or induce ciliary arrests, and thereby shift larvae upwards or downwards in the water column (*Conzelmann et al., 2011*, *2013*; *Tosches et al., 2014*). Many neuropeptides are expressed in sensory-neurosecretory cells, suggesting that sensory cues can influence larval swimming by neuroendocrine signalling (*Conzelmann et al., 2013*; *Tessmar-Raible et al., 2007*). The serotonergic and cholinergic ciliomotor neurons are also postsynaptic to functionally distinct sensory neuron pathways that can trigger ciliary arrests or induce swimming by synaptic signalling (unpublished results).

The ciliomotor system we describe here represents a novel pacemaker motor circuit that forms a stop-and-go system for ciliary swimming (*Figure 9*). Remarkably, some of the neurons in this circuit (Loop and Ser-tr1 neurons) span twice the entire body length to coordinately regulate segmental cilia. Furthermore, many cells have a biaxonal morphology, a feature that is very rare in *Platynereis* neurons belonging to other circuits (*Randel et al., 2014*; *Shahidi et al., 2015*). Comparative morphological studies using histological techniques suggest that similar ciliomotor systems, sometimes with large biaxonal neurons (*Temereva and Wanninger, 2012*), are widespread in ciliated larvae. Serotonergic neurons innervating ciliary bands have been described in several groups (*Hay-Schmidt, 2000*), including annelid (*Voronezhskaya et al., 2003*), mollusc (*Friedrich et al., 2002*; *Kempf et al., 1997*), phoronid (*Temereva and Wanninger, 2012*), and bryozoan larvae (*Wanninger et al., 2005*), and likely represent an ancestral character in lophotrochozoans (*Wanninger et al., 2005*). Serotonergic innervation of ciliary bands is also present in deuterostomes, including echinoderm and hemichordate larvae (*Hay-Schmidt, 2000*). Likewise, catecholamine-containing processes innervate ciliary bands in mollusc (*Croll et al., 1997*), phoronid (*Hay-Schmidt, 1990b*), echinoderm, hemichordate (*Dautov and Nezlin, 1992*; *Nezlin, 2000*), and nemertean larvae (*Hay-Schmidt, 1990a*). Cholinergic nerves also often associate with ciliary bands in echinoderm, hemichordate (*Dautov and Nezlin, 1992*; *Nezlin, 2000*), and mollusc larvae (*Raineri, 1995*).

One intriguing (although controversial) scenario is that ciliated larvae and their nervous systems, including the apical sensory organ (*Marlow et al., 2014*) and the ciliomotor systems, are homologous across bilaterians or even eumetazoans. If this is the case, this could mean that the *Platynereis* ciliomotor system represents an ancient system that evolved close to the origin of the nervous system (*Jékely, 2011*). The internal coordination of multiciliated surfaces may thus have been one of the early functions of nervous systems (*Jékely et al., 2015*). In-depth comparative studies are needed to understand how larval locomotor systems are related and whether ciliary coordination has a common origin or evolved multiple times independently.

## Materials and methods

### Animal culture and behavioural experiments

*Platynereis dumerilii* larvae were reared at 18°C in a 16 hr light 8 hr dark cycle until the behavioural experiments. Occasionally animals were kept overnight at 10°C to slow down growth. Larvae were raised to sexual maturity according to established breeding procedures (*Hauenschild and Fischer, 1969*). We found that nectochaete larvae showed very similar ciliary activity to trochophore larvae, but younger larvae displayed fewer muscle contractions. For this reason, we often used younger animals for recordings. Experiments were conducted at 22°C, most often between 36 and 52 hr-post-fertilisation.

### In situ hybridization and immunohistochemistry

Whole mount in situ hybridization, and confocal imaging of the animals were performed as described previously (*Asadulina et al., 2012*). The custom-made *Platynereis* allatotropin antibody was described in *Shahidi et al., 2015*. Immunostainings were done as described before (*Conzelmann and Jékely, 2012*), with the following modifications. Larvae were fixed with 4% formaldehyde in PTW (PBS + 0.1% Tween-20) for 15 min at room temperature and rinsed three times with PTW. Larvae were blocked in PTWST (PTW, 5% sheep serum, 0.1% Triton-X 100) at room

temperature, incubated overnight in 1:250 rabbit anti-HA antibody (rabbit, RRID: AB_1549585) and 1:250 anti acetylated-tubulin antibody (mouse, RRID: AB_477585) in PTWST. Larvae were washed five times for 20–60 min in PTW, blocked for 1 hr in PTWST at room temperature and incubated for 2 hr at room temperature in anti-mouse Alexa Fluor-488 1:250 (goat, RRID: AB_2534069) and anti-rabbit Alexa Fluor-633 1:250 (goat, RRID: AB_2535731), in PTWST. Larvae were washed five times for 20–60 min in PTW and transferred gradually to 97% 2,2'-thiodiethanol (TDE) (166782, Sigma-Aldrich, St. Louis, USA) in steps of 25% TDE/PTW dilutions. For the whole-mount in situ and immunostaining samples we generated projections using Imaris 8.0.2 (Bitplane, Zürich).

## Connectome reconstruction, neuron and network visualisation

Neuron reconstruction, visualization, and synapse annotation were done in Catmaid (*Schneider-Mizell et al., 2016*; *Saalfeld et al., 2009*) as described before (*Randel et al., 2015*; *Shahidi et al., 2015*). The ciliomotor neurons described here include all neurons that form at least five synapses combined on any of the multiciliated cells in the whole larval body. Six neurite fragments could not be linked to a cell body, these were omitted from the analyses.

Neurons were named based on transmitter content or morphology and position, (e.g., dorsal ciliomotor, cMN$^d$, ventral left ciliomotor, cMN$^{vl}$). The ciliated troch cells were named following the morphological literature. We numbered prototroch cells based on their position (e.g. 2 o'clock) and whether they belong to the anterior or posterior prototroch ring (e.g., proto$^{5P}$ is the posterior prototroch cell at 5 o'clock position in the ring). We grouped paratroch cells from the same larval segment into four clusters, based on their position (e.g., para$^{IIIvl1}$ is a paratroch cell in the third segment in the ventral-left cluster). Cells are labelled based on body side of their soma (left [l] or right [r]).

Network analyses were done in Catmaid (RRID:SCR_006278) and Gephi 0.8.2 (RRID:SCR_004293). Modules were detected in Gephi with an algorithm described in (*Blondel et al., 2008*) with randomization on, using edge weights and a resolution of 1.4. Force-field-based clustering was performed using the Force Atlas 2 Plugin.

Cable length was measured in Catmaid using skeleton smoothing with Gaussian convolution with a sigma of 6 μm.

## Imaging

Ciliary beating and ciliary arrests were imaged with a Zeiss Axioimager microscope (Carl Zeiss, Jena, Germany) and a DMK 21BF04 camera (The Imaging Source, Bremen, Germany) at 60 frames per second. Experiments were done at room temperature in filtered natural seawater. Larvae were immobilized between a slide and a coverslip spaced with adhesive tape. Calcium-imaging experiments were performed with GCaMP6s (*Chen et al., 2013*) mRNAs (1000 μg/μL) injected in one-cell stage as described previously (*Randel et al., 2014*). Injected larvae were mounted following the same protocol as ciliary beating experiments. Confocal images were taken on an Olympus Fluoview-1200 (with a UPLSAPO 60X water-immersion objective, NA 1.2), and a Leica TCS SP8 (with a 40X water-immersion objective, NA 1.1) confocal microscopes. Movies were acquired with a temporal resolution between 0.8 and 45 Hz. Responses to drugs were imaged 5–20 min after application to the bath and without any change of imaging settings.

## Laser ablation

Laser ablation was performed on the Olympus FV1200 confocal microscope equipped with a SIM scanner (Olympus Corporation, Tokyo, Japan), as described in (*Randel et al., 2014*). Larvae were immobilized between a slide and a coverslip. A 351 nm pulsed laser (Teem Photonics, Grenoble, France) at 90–95% power was used, coupled via air to the SIM scanner for controlled ablation in a region of interest (ROI). 95% corresponds to a beam power of 297.5 μwatts as measured with a microscope slide power sensor (S170C; Thorlabs, Newton, USA).

## Photoactivation

PAmCherry1 (Addgene plasmid 31928) was cloned into a *Platynereis* expression vector (pUC57-T7-RPP2-(NLS)pPAmCherry1) (*Randel et al., 2014*) with an NLS-tag before the N-terminus. PAmCherry1 RNA was synthesized in vitro and co-injected with GCaMP6s RNA. Photoactivation in a region of interest was performed with a 405 nm laser (0.2% power) with 2–3 200 msec pulses using

the SIM scanner of an Olympus FV12000 confocal microscope. Photo-activated larvae were recovered from the slide, fixed and processed for immunostaining with serotonin (rabbit, RRID: AB_572263) and acetylated tubulin (mouse, RRID: AB_477585) antibodies.

## Data analysis and image registration

GCaMP6s movies were analysed with FIJI (*Schindelin et al., 2012*) (RRID:SCR_002285) and a custom Python script, as described in (*Gühmann et al., 2015*) with modifications (*Verasztó, 2017*). The same ROI was used to quantify fluorescence before and after drug application. Data are presented as $\Delta F/F_0$. For the calculation of the normalized $\Delta F/F_0$ with a time-dependent baseline, $F_0$ was set as the minimum standard deviation of fluorescence in a 10-frame-window in every experiment. Videos were motion-corrected in FIJI with moco (*Dubbs et al., 2016*) and Descriptor based registration (https://github.com/fiji/Descriptor_based_registration). Correlation analyses and fast Fourier transformation to compute the discrete Fourier transform of activation sequences were done using FIJI and Python (*Verasztó, 2017*; a copy is archived at https://github.com/elifesciences-publications/Veraszto_et_al_2017). We only analysed segments of videos without movement artefacts. The ROI was defined manually and was correlated with every pixel of the time-series. Finally, a single image was created with the Pearson correlation coefficients and a $[-1, 1]$ heatmap plot with two colours.

## Acknowledgements

We thank Sara Mendes for cloning the palmitoylation tag into the reporter vector. We thank Markus Conzelmann for generating antibodies, and Christian Liebig for help with microscopy. The research leading to these results received funding from the European Research Council under the European Union's Seventh Framework Programme (FP7/2007-2013)/European Research Council Grant Agreement 260821. This project is supported by the Marie Curie ITN 'Neptune', GA 317172, funded under the FP7, PEOPLE Work Programme of the European Commission. This project is supported by the DFG - Deutsche Forschungsgemeinschaft (Reference no. JE 777/3–1).

## Additional information

### Funding

| Funder | Grant reference number | Author |
| --- | --- | --- |
| Deutsche Forschungsgemeinschaft | 777/3-1 | Gáspár Jékely |
| Max-Planck-Gesellschaft | Open-access funding | Gáspár Jékely |
| European Commission | GA 317172 | Gáspár Jékely |

The funders had no role in study design, data collection and interpretation, or the decision to submit the work for publication.

### Author contributions

CV, Conceptualization, Data curation, Software, Formal analysis, Validation, Investigation, Visualization, Methodology, Writing—review and editing; NU, LAB-C, Investigation, Visualization, Methodology, Writing—review and editing; AP, Investigation, Visualization, Methodology; EAW, Investigation, Methodology, Writing—review and editing; RS, Resources, Investigation, Visualization, Methodology, Writing—review and editing; GJ, Conceptualization, Data curation, Formal analysis, Supervision, Funding acquisition, Validation, Investigation, Visualization, Writing—original draft, Project administration, Writing—review and editing

### Author ORCIDs

Csaba Verasztó, http://orcid.org/0000-0001-6295-7148
Luis A Bezares-Calderón, http://orcid.org/0000-0001-6678-6876
Gáspár Jékely, http://orcid.org/0000-0001-8496-9836

## Additional files

### Supplementary files

• Supplementary file 1. Synaptic connectivity matrix of ciliomotor neurons and multiciliated cells.

### Major datasets

The following previously published datasets were used:

| Author(s) | Year | Dataset title | Dataset URL | Database, license, and accessibility information |
|---|---|---|---|---|
| Verasztó and Jékely | 2017 | Neuronal morphologies of all ciliomotor neurons reconstructed from a full-body serial TEM dataset of a Platynereis nectochaete larva | https://doi.org/10.13021/G8SQ34 | Publicly available at NeuroMorpho |

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
