## [Decision Letter]

Thank you for submitting your article "Ciliomotor circuitry underlying whole-body coordination of ciliary activity in the *Platynereis* larva" for consideration by *eLife*. Your article has been reviewed by two peer reviewers, and the evaluation has been overseen by Piali Sengupta as the Reviewing Editor and Eve Marder as the Senior Editor. The reviewers have opted to remain anonymous.

The reviewers have discussed the reviews with one another and the Reviewing Editor has drafted this decision to help you prepare a revised submission.

Summary:

This well-written and exciting paper describes a partial connectome for the larva of the annelid *Platynereis*. Locomotion in ciliated animals has long been interesting from a biomechanics point of view, but whole-body coordination of ciliary movements requires internal circuitry that has long been mysterious. This paper reconstructs this motor circuitry through reconstruction of serial electron micrographs, identifying all multiciliated cells as well as all the ciliomotor neurons presynaptic to them. Such circuitry has long been studied in more conventional motor systems, and understanding how such circuitry might be adapted to such a different swimmer is of fundamental interest and importance. The authors then characterize the neurochemistry of these neurons, identifying cholinergic, serotonergic and catecholaminergic/peptidergic cells. Finally, using whole-animal calcium imaging, the authors functionally characterize the motor circuitry by examining temporal correlations between neurons and their synaptic targets. Basic puzzles like the locus of the rhythm generator remain to be discovered, but the authors are very clear about outstanding questions in the field.

Minor points:

Textual clarification: in general, the paper seems to assume that the identified neurotransmitters/modulators act at synapses, but amines and peptides in particular are known to often act extrasynaptically and at a distance between neurons that may or may not be synaptically connected. The authors should discuss this issue in the text.

---

## [Author Response]

*Minor points:*

*Textual clarification: in general, the paper seems to assume that the identified neurotransmitters/modulators act at synapses, but amines and peptides in particular are known to often act extrasynaptically and at a distance between neurons that may or may not be synaptically connected. The authors should discuss this issue in the text.*

We have now included the possibility of extrasynaptic signalling in the Discussion.

“The monoamines and neuropeptides we mapped to the ciliomotor neurons may act at synapses or extrasynaptically (Bentley et al. 2016). Understanding the details of signalling will require the identification of the receptor expressing cells. We recently characterised several neuropeptide receptors, adrenergic receptors and other monoamine receptors in *Platynereis* (Bauknecht and Jékely 2017; Bauknecht and Jékely 2015). We also developed a method to dissect chemical connectivity of neuropeptide signalling with cellular resolution (Williams et al. 2017). These approaches will help to dissect the mechanisms of rhythm generation and the modulation of the rhythm and ciliary activity.”